# A Structure-Based Iterative Closest Point Using Anderson Acceleration for Point Clouds with Low Overlap

**DOI:** 10.3390/s23042049

**Published:** 2023-02-11

**Authors:** Chao Zeng, Xiaomei Chen, Yongtian Zhang, Kun Gao

**Affiliations:** 1School of Optics and Photonics, Beijing Institute of Technology, Beijing 100081, China; 2MOE Key Laboratory of Optoelectronic Imaging Technology and System, Beijing Institute of Technology, Beijing 100081, China

**Keywords:** registration, point clouds, overlap, Anderson acceleration

## Abstract

The traditional point-cloud registration algorithms require large overlap between scans, which imposes strict constrains on data acquisition. To facilitate registration, the user has to strategically position or move the scanner to ensure proper overlap. In this work, we design a method of feature extraction based on high-level information to establish structure correspondences and an optimization problem. And we rewrite it as a fixed-point problem and apply the Lie algebra to parameterize the transform matrix. To speed up convergence, we introduce Anderson acceleration, an approach enhanced by heuristics. Our model attends to the structural features of the region of overlap instead of the correspondence between points. The experimental results show the proposed ICP method is robust, has a high accuracy of registration on point clouds with low overlap on a laser datasets, and achieves a computational time that is competitive with that of prevalent methods.

## 1. Introduction

Rigid point-cloud registration consists of finding the best transformation between point clouds, and plays a key role in robotics and computer vision. Its wide application has rendered it ubiquitous, such as in simultaneous localization and mapping (SLAM) [1,2], 3D reconstruction [3], and motion estimation [4]. At first glance, its performance on various benchmarks gives the impression that it can solve the problem of registration. However, it has now been noted that this is not the case owing to the leniency of the evaluation protocols used, where the benchmarks used in recent literature have considered the overlap between only pairs of point clouds ≥80% and reached nearly ≥90% matching recall [5,6]. The registration-related performance of prevalent methods deteriorate rapidly in practice when the input data contain a region with a low overlap with the template data [7]. Moreover, many approaches are not capable of being generalized to large-scale empirical 3D point clouds owing to their different densities and the characteristics of LiDAR sensors [8].

The classic Iterative Closest Points (ICP) [9] algorithm alternates between querying the closest point in the target set and minimizing the distance between corresponding points, and guarantees convergence to an optimal alignment. However, the major shortcoming of the ICP when used for registration is that it can converge to only a local optimization near the initialization. The search for corresponding points is time consuming when large-scale point clouds need to be processed, and the differences in their densities, missing data, and regions of low overlap in laser scans lead to poor registration-related performance [10], as illustrated in the example shown in Figure 1.

In this work, we address the registration of pairs of point clouds with a low overlap as captured by LiDAR from real-world scenes. Point clouds scanned by different LiDAR-based methods vary greatly. Thus, different approaches that rely on a sufficient amount of overlapping and descriptive features cannot guarantee the performance expected of them. In fact, man-made scenes like building interiors and exteriors comprising mainly planar and edge-related structures are universal in the real-world. We think that edge-related and planar features along with their inter-relations reveal high-level global characteristics that provide a sufficient amount of information for point cloud registration. However, currently used plane-/line-based methods have unsatisfactory robustness and a high time cost [6,10,11].

Key to our approach here is a combination of planar and edge-related features as well as the solution of a fixed-point problem. Specifically, we extract planar and edge-related features from a raw point cloud captured by the mechanical LiDAR and the solid-state LiDAR. And then we establish structure-level correspondences between them and formulate the optimization problem. Note that the non-linear optimization problem can be rewritten as a fixed-point equation, which can be solved by Anderson acceleration. Instead of standard iteration usually used in ICP, this method is an iterative procedure for finding a fixed point of contractive mapping. Nevertheless, the value computed by Anderson Acceleration is an affine combination of rotation matrices. To address it, we apply the Lie algebra to parameterize the transformation.

We provide benchmark datasets in this paper that are obtained through scans of a set of indoor and outdoor scenes with a low overlap. We use RealSense L515 to scan indoor scenes as it has a high angular resolution and a high density of points [12]. The outdoor dataset used here consisted of KITTI data captured by using the Volodyne HDL-64E LiDAR sensor [13]. The performance of the method of registration was simply measured based on the percentage of successful registration scans, and the differences between the estimated transformation and the ground truth.

## 2. Related Work

Current methods used to align two input point clouds can be classified according to the approaches that they apply to find the correspondences that are used for motion estimation.

### 2.1. Correspondences Based on 3D Features

Algorithms in this category are prevalent in point cloud registration. They focus on using/defining different features of local or global points to establish putative correspondences between the key points extracted from raw point clouds [14], and then use closed-form solutions or robust optimization algorithms to realize the transformation [15].

SIFT3D [16] is an extension of the original 2D SIFT detector in which the difference between Gaussian pyramids is used to find the key points. Steder et al. [17] exploit the NARF detector to select points at which the surface is stable and that features a sufficient amount of change in the local neighborhood. It is also important to identify general and robust 3D feature descriptors. The point feature histogram (PFH) [18] and the fast point feature histogram (FPFH) [19] represent relationships between a given point and its neighbors. Similarly, some global shape descriptors have been proposed to align point clouds, such as Hough transform descriptors [20], the spherical entropy image [21], and viewpoint descriptors [22].

By contrast, recent studies have used data-driven deep neural networks to learn feature descriptors from large-scale datasets. Zeng et al. [5] propose a 3D convolution neural network (3DMatch) that uses the local volumetric region around a given key point to compute feature descriptors for it. Building on sparse convolutions, FCGF [23] achieves a performance similar to the best patch-based descriptors [24], while being orders of magnitude faster. D3feat [25] leverages a 3D fully convolution network to predict both a detection score and a description feature for each point. SpinNet [26] introduces a Spatial Point Transformer and 3D cylindrical convolution to extract local features which are rotationally invariant. Recently, Predator [7] utilizes attention mechanisms to aggregate contextual information to learn more discriminative feature descriptor. To simplify the architecture, REGTR [27] and JAR-Net [28] directly predict clean correspondences using multiple transformer layers.

Many effective strategies have been developed to reject outliers in the correspondences. RANSAC [29] and its variants are commonly used to this end. In the recent, graph-based methods have been widely applied to model fitting. Lin et al. [30] propose a method based on co-clustering on bipartite graphs to estimate model in data contaminated with outliers and noise.

The above-mentioned methods have focused on individual local patches, where this significantly increases computational cost. Moreover, they are usually tailored to specific tasks, and thus are not sufficiently flexible and descriptive for complicated and novel scenarios.

### 2.2. Handcrafted Registration Methods

The well-known iterative closest point [9] method is considered a milestone in point cloud registration. It uses iterative methods to establish correspondences by using the point-to-point distance and updating the transformation. However, motion estimation by it is sub-optimal owing to a poor initial transformation and the large amount of requisite calculation [22].

Several studies have focused on overcoming the shortcomings of ICP. Lecoy et al. [31] propose a method to compare variants of ICP, and use it to examine a combination of variants optimized for speed. Billings et al. [32] introduce the IMLP to robustly solve the problem of rigid-body alignment. The PLICP [10] uses the point-to-line metric and an exact closed form for minimizing it. It is more precise and requires fewer iterations than the IMLP. The point-to-plane ICP [33] minimizes the orthogonal distance between points in one point set and the corresponding local planes in the other to estimate the motion transform. Motion estimation by using the plane-to-plane distance is more efficient for human-made scenes than point-to-point and point-to-plane correspondences [11]. Nevertheless, most ICP variant still require relatively good initialization to avoid converging to bad local minima. A notable exception, the Go-ICP [34] method is based on a branch-and-bound scheme that searches the entire 3D motion space.

### 2.3. Deep Point-Cloud Registration

Instead of combining learned feature descriptors with robust optimization at inference time, some works incorporate the entire pose estimation into the training pipeline. PointNetLK [35] combines a PointNet-based global feature descriptor [36] with the the Lucas & Kanade (LK) algorithm and estimates relative transformation. DeepVCP [37] innovatively generates corresponding points based on learned matching probabilities among a group of candidates. Deep Closest Point (DCP) [38] proposes a learned version of Iterative Closest Point (ICP) [9] and utilizes soft correspondences on learned pointwise features to compute the rigid transform. Pointdsc [39] formulates a nonlocal feature aggregation module and a differentiable spectral matching module to prune outlier correspondences. OMNet [40] learns overlapping masks to reject non-overlapping regions, which converts the partial-to-partial registration to the registration of the same shape.

Many methods achieve the expected score on the training set, but them have relatively poor generalization ability across the unseen dataset. Moreover, a few recent studies have proposed methods that either require computing the density of points or rely on the region of a high overlap between point clouds, and thus are time consuming.

## 3. Materials and Methods

### 3.1. Problem Formulation and Classic ICP Revisited

Let the two sets of points $P=\left\{p_{1},\cdots ,p_{M}\right\}$ and $Q=\left\{q_{1},\cdots ,q_{N}\right\}$, which represent point coordinates in $R^{3}$, be the source point set and the target point set, respectively. The goal of point cloud registration is to estimate a rigid transformation T (represented by using a rotation matrix $R\in R^{3\times 3}$ and a translation vector $t\in R^{3}$) on *P* to align it with *Q*. This minimizes the following $l_{2}$-norm error *E*: (1)$$E\left(R,t\right)=\sum_{i=1}^{M}e_{i}\left(R,t\right)+I_{SO(d)}\left(R\right),$$
where $e_{i}\left(R,t\right)={∥Rp_{i}+t-q_{i}∥}^{2}$ is the per-point residual error between the transformed source point $Rp_{i}+t$ and the closest target point $q_{i}$, and $I_{SO(d)}\left(\cdot \right)$ is an indicator function that requires that R be a rotation matrix: (2)$$I_{SO(d)}\left(R\right)=\left\{\begin{array}{cc} 0, & \;\mathrm{if}\;R^{T}R=I\;\mathrm{and}\;\det\left(R\right)=1\\ +\infty & \;\mathrm{others}. \end{array}\right.$$

Equations (Equation 1) and (Equation 2) represent a well-known chicken-and-egg problem that cannot be solved trivially. The ICP algorithm is frequently used to solve the problem of registration.

Given an initial guess for a rigid transformation $T^{0}$, the ICP algorithm repeats the following two steps:(1)Find the corresponding closest point $q_{i}^{\left(k\right)}\in Q$ for each point $p_{i}^{\left(k\right)}\in P$ based on the last iterative transformation $T^{\left(k\right)}$.(2)Update the transformation by minimizing the $l_{2}$-norm error *E* between the corresponding points, and render the result as the transformation $T^{\left(k+1\right)}$.
Such an iterative method can ensure that the transformation matrix can converge to a local optimum. Despite its simplicity, the classical ICP converges slowly to a local minimum because of its linear convergence and search speed. In addition, in case of a low overlap between the source and target point sets, ICP generates an erroneous alignment. Point clouds with different ratios of overlap are shown in Figure 2.

### 3.2. Feature Point Extraction

LiDAR measures the distance to a given object by using a laser and calculating the distance traveled by the beam. Current sensors based on LiDAR can be classified into mechanical multi-line spinning LiDAR sensors and solid-state LiDAR sensors. The scanning of point clouds by using different LiDAR sensors yields discrepancies in the results. Solid-state LiDAR sensors usually have a small field of view (FoV) and an irregular scanning pattern that lead to a high angular resolution and dense point clouds in the scene. On the contrary, mechanical LiDAR has a line-scan pattern that enables it to obtain a sparse point cloud by spinning the laser array at a high frequency.

In general, registering the raw point clouds captured by LiDAR sensors is too computationally burdensome. In particular in case of scenes with a small overlap, the iterations are prone to converging to the local optimum. The real-world environment typically consists of a line–plane geometry. Therefore, we select feature points that are along sharp edges and on planar surface patches to reduce the amount of requisite computation and eliminate noise. The distortion in motion is eliminated from the data by using linear interpolation.

For the scanning of point clouds by multi-line spinning LiDAR, let $p_{i}$ be a point, $p_{i}\in S$, and let S be the set of consecutive points of $p_{i}$ on the same scanning line. To differentiate between edge and planar points, we compute the local smoothness of the candidate point $p_{i}$ by searching its neighboring points:(3)$$\varrho_{i}=\frac{1}{\lambda^{2}}\sum_{p_{j}\in S_{i},j\neq i}\left(∥p_{j}∥-∥p_{i}∥\right)$$
with
(4)$$S_{i}=\left\{p_{j}\in S|j\in \left[i-\lambda ,i+\lambda \right]\right\}$$
where λ is the radius of search around point $p_{i}$. The points in S are sorted based on their local smoothness. The value of $\varrho_{i}$ of points along the edges is higher than that of points on the plane.

To select feature point clouds from the evenly distributed raw point set, we separate a scan line into four sub-regions that can provide a maximum of two edge points and five planar points. A point with local smoothness $\varrho_{i}$ can be selected as an edge or a planar point only if $\varrho_{i}$ is larger or smaller than a pre-defined threshold and the number of points selected is smaller than the maximum number possible.

To scan the point clouds by using the solid-state LiDAR, we need different strategies to extract the feature points owing to the irregular pattern and density of the point clouds. Random sample consensus (RANSAC) can be applied to extract the features. The classic RANSAC collects point clouds *P* by scanning a scene, including edge-related and planar features. We select a random sample subset *S* of the original data, and choose *n* points to form an initial model *M* with the objective function *C*. The points in the complementary set fit model *M*, with the initial points forming the set of inliers. If the number of inliers exceeds the standard value, we use the set of inliers to construct the new model $M^{*}$ by using the least-squares method. This process is continued until the maximum interior point set has been obtained, at which time the point set is considered to satisfy the requirements of the model.

Because random point sampling reduces the efficiency of the algorithm, the subset of random samples *S* is initially judged. In case the initial points can form a planar or an edge-based model, the number of point clouds in the sphere of radius $r_{0}$, centered at the initial point, that are needed to form the model are assessed. If this number is insufficient, the initial point is randomly selected again.

The edge-based and planar points are selected from the point cloud as shown in Figure 3.

### 3.3. Error Model

As mentioned above, registering raw point clouds is inefficient, and is sensitive to noise and outliers. We thus use matching edge-related and planar points from raw point clouds in the feature space. We assume that $E_{s}$ and $E_{t}$ are the sets of all edge-related points in the source and the target point sets, respectively. To increase the search speed, the target edge-related point set $E_{t}$ is built by using the KD tree.

Figure 3a shows the correspondence between edge-related features in the source and target point clouds. For each edge-related point $p_{k}^{E}\in E_{s}$, we obtain the transform ${\hat{p}}_{k}^{E}=R^{0}p_{k}^{\varepsilon }+t^{0}$ based on a guess regarding the initial transformation $t^{0}$. We can search two nearest edge-related feature points $p_{1}^{E}$ and $p_{1}^{E}$ from the target edge-related feature set $E_{t}$. Thus, the point-to-edge distance $d_{\varepsilon }\left({\hat{p}}_{k}^{E}\right)$ between ${\hat{p}}_{k}^{E}$, and the edge crossing $p_{1}^{E}$ and $p_{2}^{E}$ is computed as: (5)$$d_{\varepsilon }\left({\hat{p}}_{k}^{E}\right)=\frac{\left|\left({\hat{p}}_{k}^{E}-p_{1}^{E}\right)\times \left({\hat{p}}_{k}^{E}-p_{2}^{E}\right)\right|}{\left|\left(p_{1}^{E}-p_{2}^{E}\right)\right|}$$
where |·| represents the modules of the vector and × is the cross-product of two vectors. Only when the point-to-edge distance is shorter than the pre-set threshold is the model optimized.

Similarly, we denote by $P_{s}$ and $P_{t}$ the set of all planar features in the source and the target point sets, respectively, in Figure 3b. For each planar feature $p_{k}^{P}\in P_{s}$, it is necessary to search three planar points nearest to its transform ${\hat{p}}_{k}^{P}=R^{0}p_{k}^{P}+t^{0}$ by estimating a plane in 3D space. We select the points $p_{1}^{P}$, $p_{2}^{P}$, and $p_{3}^{P}$ from the target planar feature set. The point-to-plane distance $d_{P}\left({\hat{p}}_{k}^{P}\right)$ between $p_{k}^{P}$, and the edge crossing $p_{1}^{P}$, $p_{2}^{P}$ and $p_{3}^{P}$ is then computed as: (6)$$d_{P}\left({\hat{p}}_{k}^{P}\right)=\frac{\left|{\left({\hat{p}}_{k}^{P}-p_{2}^{P}\right)}^{T}\cdot \left(p_{1}^{P}-p_{2}^{P}\right)\times \left(p_{1}^{P}-p_{3}^{P}\right)\right|}{\left|\left(p_{1}^{P}-p_{2}^{P}\right)\times \left(p_{1}^{P}-p_{3}^{P}\right)\right|}$$

Only when the distance is shorter than the pre-set threshold is it considered. The final rigid transformation can be estimated by minimizing the point-to-edge distance and the point-to-plane distance: (7)$$\arg\max_{R,t}\frac{1}{N_{E}}\sum_{i=1}^{N_{E}}d_{\varepsilon }\left({\hat{p}}_{k}^{\varepsilon }\right)+\frac{1}{N_{P}}\sum_{i=1}^{N_{P}}d_{P}\left({\hat{p}}_{k}^{P}\right)+I_{SO(d)}\left(R\right)$$

As with ICP, we can use an iterative approach and the Gauss–Newton optimization to solve the non-linear optimization problem. However, the cost and efficiency of the calculations need to be considered.

### 3.4. Anderson Acceleration for Fixed-Point Problem

Anderson Acceleration: By assuming a fixed-point iteration $g^{\left(k+1\right)}=G\left(g^{\left(k\right)}\right)$, we define its residual function f(g)=G(g)−g, and $f^{\left(k\right)}=G\left(g^{\left(k\right)}\right)-g^{\left(k\right)}$. According to its definition, a fixed point $g^{*}$ of the mapping G(·) satisfies $f\left(g^{*}\right)=0$, and can be solved by fixed-point iteration. To accelerate convergence, Anderson acceleration utilizes the latest iteration $g^{\left(k\right)}$ as well as the preceding *m* iterations $g^{\left(k-m\right)},g^{\left(k-m+1\right)},\cdots ,g^{\left(k-1\right)}$ to derive a new iteration $g^{\left(k+1\right)}$ that converges more quickly to a fixed point [41]: (8)$$g^{\left(k+1\right)}=G\left(g^{\left(k\right)}\right)-\sum_{j=1}^{m}\theta_{j}^{*}\left(G\left(g^{\left(k-j+1\right)}\right)-G\left(g^{\left(k-j\right)}\right)\right)$$
where $\left(\theta_{1}^{*},\cdots ,\theta_{m}^{*}\right)$ is solution to the following linear least-squares problem: (9)$$\left(\theta_{1}^{*},\cdots ,\theta_{m}^{*}\right)=\arg\min{∥f^{\left(k\right)}-\sum_{j=1}^{m}\theta_{j}\left(f^{\left(k-j+1\right)}-f^{\left(k-j\right)}\right)∥}^{2}$$

We note several properties of the ICP that make it computationally expensive and challenging to find the rigid transformation of Equation (Equation 7) in order to accelerate convergence by using higher-order methods.

We write our target function, explained in Section 2.3, as a fixed-point iteration of the transformation variables R and t: (10)$$\left(R^{\left(k+1\right)},t^{\left(k+1\right)}\right)=G\left(R^{\left(k\right)},t^{\left(k\right)}\right)$$
where
(11)$$G\left(R^{\left(k\right)},t^{\left(k\right)}\right)=\arg\max_{R,t}\frac{1}{N_{\varepsilon }}\sum_{i=1}^{N_{E}}d_{\varepsilon }\left(R^{\left(k\right)}p_{k}^{E}+t^{\left(k\right)}\right)+\frac{1}{N_{P}}\sum_{i=1}^{N_{P}}d_{P}\left(R^{\left(k\right)}p_{k}^{P}+t^{\left(k\right)}\right)+I_{SO(d)}\left(R\right)$$

$d_{\varepsilon }$ denotes the distance between $R^{\left(k\right)}p_{k}^{E}$ and the closest edge to it, and $d_{P}$ denotes the distance between $R^{\left(k\right)}p_{k}^{P}$ and the plane closest to it.

However, we cannot directly apply Anderson acceleration to the mapping G(·) because the new value of R computed by Anderson acceleration is an affine combination of rotation matrices, where $R^{T}R\neq I$ and det(R)≠1. We can parameterize a rigid transformation (R,t) by using another set of variables X. Equation (Equation 9) can then be re-written as: (12)$$X^{\left(k+1\right)}=\bar{G}\left(X^{\left(k\right)}\right)$$

When applying Anderson acceleration, we can then parameterize the transformation to obtain $X^{\left(k\right)}$. When computing the optimization function, we can recover the rotation matrix $R^{\left(k\right)}$ and the translation vector $t^{\left(k\right)}$ from the variable $X^{\left(k\right)}$.

The parameterization of a rigid transformation involves concatenating the translation vector and the parameterized rotation matrix, where this can be represented as Euler angles or unit quaternions in $R^{4}$ [42]. However, the Euler angle has singularities called gimbal lock and the affine combination of unit quaternions does not result in a unit quaternion. Nonetheless, all rigid transformations in $R^{4}$ form the special Euclidean group SE(3), which is a Lie group that gives rise to a Lie algebra se(3) [43], and Lie algebras are closed under addition. Thus, we define X=[ρ,ϕ]∈se(3) and recover the transformation matrix from it: (13)$$\left[\begin{array}{cc} R & t\\ 0^{T} & 1 \end{array}\right]=\exp\left(X^{\wedge }\right)$$
where $X^{\wedge }$ converts a 6D vector into a 4×4 matrix by
(14)$$X^{\wedge }=\left[\begin{array}{cc} {\left[\rho \right]}_{\times } & \phi \\ 0_{1\times 3} & 1 \end{array}\right]$$
where ${\left[\cdot \right]}_{\times }$ is the skew matrix of a 3D vector. We can then execute Anderson acceleration on the Lie algebra of the transformation matrices. To avoid the instability and stagnation of Anderson acceleration, we accept the accelerated value as the new value only if it reduces the target function compared with the value in the previous iteration [44].

We can use the left perturbation scheme and apply increment on the Lie group to solve the non-linear optimization problem (Equation (Equation 7)) by using the Gauss–Newton algorithm. The left perturbation model can be formulated as follows: (15)$$J_{p}=\frac{\partial Tp}{\partial \delta X}=\lim_{\delta X\to 0}\frac{\left(\exp\left(\delta X^{\wedge }\right)\cdot Tp-Tp\right)}{\delta X}=\left[\begin{array}{cc} I_{3\times 3} & -{\left[Tp\right]}_{\times }\\ 0_{1\times 3} & 0_{1\times 3} \end{array}\right]$$

Note that ${\left[Tp\right]}_{\times }$ transforms the 4D point expression {x,y,z,1} into the 3D point expression {x,y,z} before calculating the skew matrix. Thus, the Jacobian matrix of the point-to-edge distance $d_{\varepsilon }\left(p_{k}^{E}\right)$ is defined by
(16)$$J_{\varepsilon }=\frac{\partial d_{\varepsilon }\left({\hat{p}}_{k}^{E}\right)}{\partial \delta X}=\frac{\partial d_{\varepsilon }\left({\hat{p}}_{k}^{E}\right)}{\partial {\hat{p}}_{k}^{E}}\frac{\partial {\hat{p}}_{k}^{E}}{\partial \delta X}=\frac{\partial d_{\varepsilon }\left({\hat{p}}_{k}^{E}\right)}{\partial {\hat{p}}_{k}^{E}}J_{p}$$

Similarly, the Jacobian matrix of the point-to-edge distance $d_{P}\left({\hat{p}}_{k}^{P}\right)$ can be written as:(17)$$J_{P}=\frac{\partial d_{P}\left({\hat{p}}_{k}^{P}\right)}{\partial {\hat{p}}_{k}^{P}}J_{p}$$

This formulation has several advantages, such as the sorting of the rotation in a singularity-free format and unconstrained optimization.

## 4. Experiments and Results

In this section, we evaluate the proposed method and justify our design-related choices on empirically acquired point clouds. We used RealSense L515 to scan indoor scenes. It had a solid-state LiDAR with a small FoV and a viewing angle of 70° × 55°. The outdoor dataset was assembled from outdoor scenes in the KITTI odometry dataset. Our method was coded in C++, and implemented on Ubuntu 18.04 and ROS Melodic.

### 4.1. RealSense L515 Data

In our experiments, we used the hand-held Intel RealSense L515 with a frequency of update of 30 Hz to scan the indoor scenes. The ground truth was provided by a VICON system.

We collect the point clouds of indoor scenes. And by means of segmentation and transformation, we add its counterpart in which we consider scan pairs with overlaps between 100 and 10%. For fairness, we made sure that the same pair of source and template point clouds with different overlapping regions were used to test. We compare our methods to recent registration methods: handcrafted methods including ICP [9], GICP [33], NDT [45] and HMRF-ICP [46], feature-based method including FPFH [19], deep point-cloud registration methods including PointNetLK [35] and Predator [7]. The dataset built from raw scans contained 1000 point cloud pairs uniformly distributed over the rates of overlap. The relative translational error (RTE), relative rotational error (RRE), and run time were used as evaluation metrics, as shown in Table 1. For registration methods that are not end-to-end, the run time includes feature extraction and transformation matrix calculation.

Our method outperformed most of the other optimization-based approaches tested, including the ICP, GICP, NDT and HMRF-ICP. And FPFH focused on the feature of local point cloud, and this led to many inaccurate correspondence pairs, especially in case of a low rate of overlap. For deep learning-based approaches, Predator had achieved satisfactory performance except runtime. Due to the use of edge-related and planar features, our method was able to register all the scanned pairs. Although some of the indoor scenes considered here contained curved surfaces, planar and edge structures still dominated them such that our method successfully registered these dense point clouds. Even with low overlap, we still established line-plane geometry correspondence accurately and easily as obvious structure features. And then use optimization methods to get registration transformation with Anderson Acceleration.

Figure 4 shows that we registered the two point sets scanned by L515 with overlaps of 20%, 60%, and 90%. Thanks to the line-plane geometry, our registration method managed to register all these scan pairs. Though some scenes partially consist of curved surfaces, the extractive feature points still provide sufficient information for a reliable registration.

Our method is capable of registering scans with different overlap. In Figure 5a, We found that filtering would reduce the accuracy of registration, so we tried our best to increase the proportion of sampling while ensuring the efficiency of the algorithm. We also tested our method to determine whether it focused on structural correspondences when the rate of overlap between the point sets decreased. We extracted test pairs by varying the completeness of the input point clouds from 100% to 10%. Remarkably, about 70% of the test pairs contained a line–plane geometry. Figure 5b shows that our method maintained a higher robustness in terms of registration than the other methods.

### 4.2. KITTI Odometry Dataset

The KITTI odometry dataset, acquired by using Velodyne-64 3D LiDAR scanners, is an outdoor dataset of sparse point clouds. It consists of 11 sequences of outdoors scans. We used the same evaluation metrics for it as for the RealSense L515 dataset. However, the KITTI odometry dataset does not have a low rate of overlap during the acquisition of the point cloud. Moreover, the point clouds were gravitationally aligned in this dataset such that the reference axis was aligned.

It is clear that the point clouds in the KITTI odometry dataset were significantly different from those in the RealSense L515 dataset because the former was mainly composed of large-scale, sparse, and partial LiDAR scans. In addition, its overlap ratio was approximately 100%, and its rotation around the *x*- and *y*-axes was roughly 0°.

As shown in Table 2 and Figure 6, compared with the performance of the ICP and Predator algorithm, our method still achieved satisfactory results of registration on it. Especially, the feature extractor and Anderson acceleration used for the fixed-point problem ensured its adequate performance in terms of computation time.

## 5. Discussion

Since the traditional correspondence-based registration algorithm requires the local feature calculation, the feature contains the information of the local point cloud, which is indeed advantageous. However, experiments show that the edge area of the point cloud and the non-overlapping region of the scans causes inaccurate correspondence, which affects the quality of registration. At the same time, calculating the local features increases the time complexity of the algorithm. This paper uses high-level structural information of raw laser scans for point correspondence, and formulate a fixed-point problem by examining correspondences between feature points, which can be solved by the Lie derivative and Anderson Acceleration.

Actually, our high-level structural feature extraction is dedicated to registering point clouds of scenes that at least partially consist of planar and edge structures. Thus, the proposed method is especially suitable for registration scans of man-made environments. For scans of vegetation and scans of individual objects that consist of curved surfaces, the extraction of plane and edge feature will probably not perform as expected.

Our work can be extended by addressing its shortcomings discussed in detail. In the one hand, our approach relies strongly on the feature detection algorithm. These steps not only influence accuracy of results of our proposed method of transformation search, but also make up to ≥50% of algorithm runtime, suggesting that using a more sophisticated method for feature detection in terms of speed and accuracy is a key step to make our method effective and applicable. In the other hand, despite the Anderson acceleration is applied, we still need iteration to converge. In future work, the deep-learning-based method can be utilized to solve the fixed-point problem.

## 6. Conclusions

In this study, we propose an approach for the pairwise registration of point clouds with a low overlap as captured by mechanical LiDAR and solid-state LiDAR. The core of our method is a combination of planar features and edge-related features as well as the formulation of a fixed-point problem, with the aim of finding the relations between the point clouds to identify their high-level characteristics and improve the speed of convergence. The main contributions of this paper are as follows:We propose a method to extract planar and edge-related feature points in data obtained from mechanical LiDAR and solid-state LiDAR.We formulate a non-linear optimization problem by examining structural correspondences between the planar and the edge-related points.Rewritten the problem as a fixed-point equation, we apply Anderson acceleration to speed up convergence and use the Lie algebra to represent a rigid transformation when solving the optimization function.

The results of simulations showed that our method achieves accurate results on scans of indoor and outdoor scenes. In future work, we plan to investigate the descriptors of feature points for scans without a line–plane structure and the solution of fixed-point problem.

## Figures and Tables

**Figure 1 sensors-23-02049-f001:**
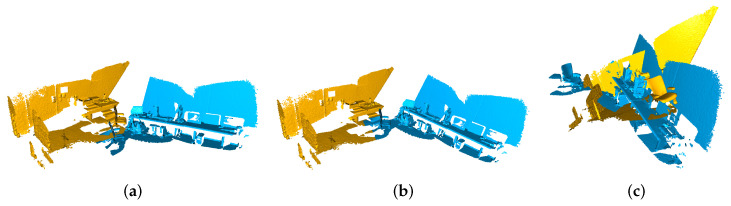
When registering point clouds with regions of low overlap, the iterative closest point method is prone to becoming trapped in a local minimum. (**a**) Input point clouds. (**b**) Ground-truth registration. (**c**) The result of the ICP.

**Figure 2 sensors-23-02049-f002:**
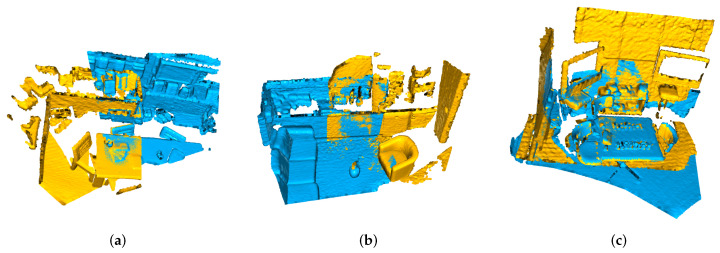
Point clouds with different overlap ratios. The overlap is computed relative to the source and target fragments. When matching point clouds with a low overlap, it is difficult to find the correct corresponding points. (**a**) overlap ratio = 0.2. (**b**) overlap ratio = 0.5. (**c**) overlap ratio = 0.8.

**Figure 3 sensors-23-02049-f003:**
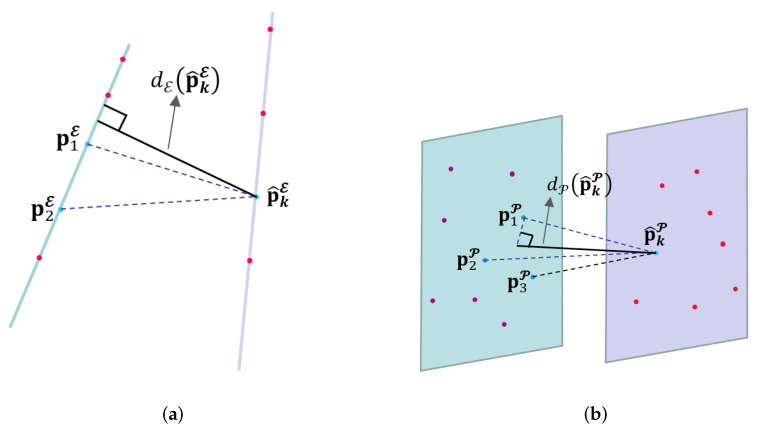
A feature in current scan and the corresponding feature points in the last scan: (**a**) edge-related feature; (**b**) planar feature.

**Figure 4 sensors-23-02049-f004:**
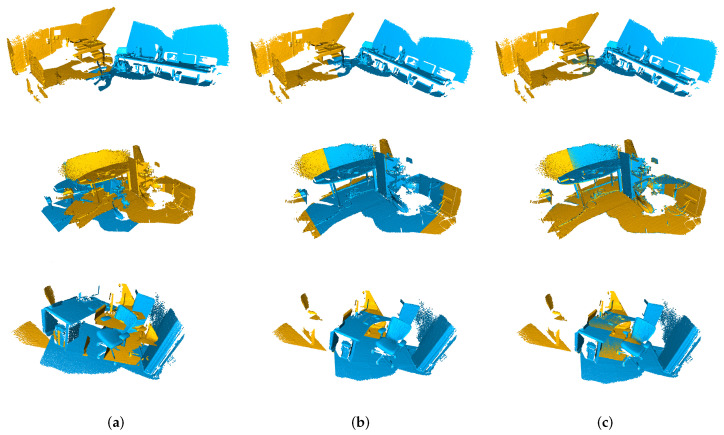
Example of results on the RealSense L515 dataset. (**a**) Input point clouds. (**b**) Ground-truth registration. (**c**) Our method.

**Figure 5 sensors-23-02049-f005:**
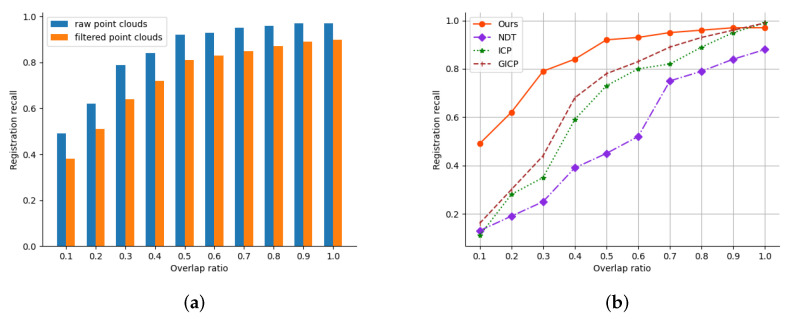
(**a**) Our method avoided failure by increasing the fraction of sampled points because dense points are beneficial for extracting structural features. (**b**) As the rate of overlap decreased, the recall of registration decreased to a certain extent, but our method was more robust than the others at a low rate of overlap.

**Figure 6 sensors-23-02049-f006:**
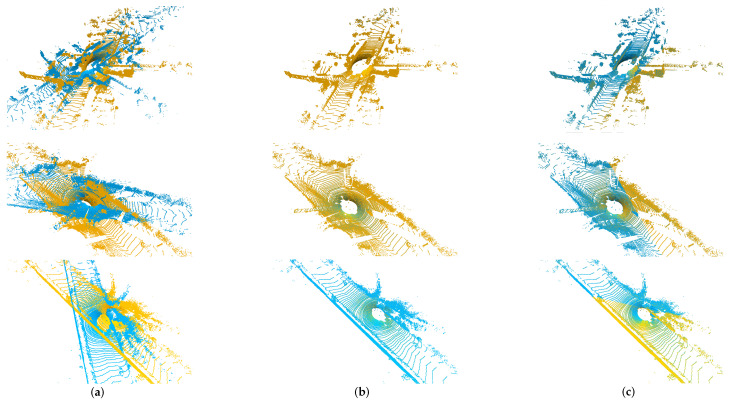
Example of results on the KITTI odometry dataset. (**a**) Input point clouds. (**b**) Ground-truth registration. (**c**) Our method.

**Table 1 sensors-23-02049-t001:** Average evaluation of our method on the indoor dataset.

	Relative Translation Error [cm]	Relative Rotation Error [°]	Run Time for Processing [s]
ICP [9]	7.32	0.42	28.71
GICP [33]	3.24	0.31	142.2
NDT [45]	38.92	12.83	23.21
HMRF-ICP [46]	6.31	0.89	13.41
FPFH [19]	9.12	1.39	16.96 + 3.22
PointNetLK [35]	12.51	2.31	21.41
Predator [7]	3.18	0.22	7.19
Our methods	3.25	0.21	2.25 + 2.31

**Table 2 sensors-23-02049-t002:** Average values of the performance metrics of our method on the KITTI odometry dataset.

	Relative Translation Error [cm]	Relative Rotation Error [°]	Run Time for Processing [s]
ICP [9]	6.29	0.16	26.81
GICP [33]	4.92	0.31	172.6
NDT [45]	29.67	12.83	32.71
HMRF-ICP [46]	8.54	1.01	11.98
FPFH [19]	11.12	1.77	14.26 + 4.51
PointNetLK [35]	14.32	1.97	18.49
Predator [7]	5.12	0.19	5.29
Our methods	5.01	0.21	1.03 + 1.37

## Data Availability

The data presented in this study are openly available from KITTI at https://www.cvlibs.net/datasets/kitti/ (accessed on 15 September 2022), reference number [13].

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
