# Peer review of "A Structure-Based Iterative Closest Point Using Anderson Acceleration for Point Clouds with Low Overlap"

_sensors, 2023, doi:10.3390/s23042049_

Round 1
Reviewer 1 Report
This paper tries to improve traditional iterative closest point (ICP) algorithm, by introducing high-level structural information for raw laser scans. The idea is interesting, but there are some serious problems:
1. The motivation and contributions are not very clear. The real contribution is "a combination of planar and edge-related features as well as the establishment of a fixed-point problem", If yes, the contribution is so limited.
2. The related work: it would be better to introduce some more recent references, such as:
(a) some outlier removal methods:
-Co-clustering on Bipartite Graphs for Robust Model Fitting, IEEE Transactions on Image Processing (TIP), 2022, 31, 6605-6620
Some Correspondence-based Methods:
-JRA-Net: Joint Representation Attention Network for Correspondence Learning, Pattern Recognition (PR), 2023
3. The experiment: The proposed method only compare a few competing methods. As we know, point cloud is a pupular topic, so it would be better to add more competing methods.
Reviewer 2 Report
overall structure of article is good and well written. article provide a good approach to speed up ICP initialization and optimization problem. However I have following concerns:
In table 1 and table 2, run time for processing is presented however it is not mentioned either time for Feature Point Extraction is included or not. kindly elaborate it.
Kindly add discussion section to discuss limitations of methods (e.g. method can be used in only such scenes where planes and lines exist) and give a general overview how these limitations will be removed in future.
kindly improved the title of article for your application environment and method to make it more precise. it seems very generalized.
Round 2
Reviewer 1 Report
The authors have considered my problems. This version is find for me.